# Effects of Different Farming Modes on *Salmo trutta fario* Growth and Intestinal Microbial Community

**DOI:** 10.3390/microorganisms12061082

**Published:** 2024-05-27

**Authors:** Zhuang-Zhuang Wang, Zhi-Tong Wang, Wan-Liang Wang, Kuan-Kuan Lei, Jian-She Zhou

**Affiliations:** 1Institute of Aquatic Sciences, Tibet Autonomous Region Academy of Agricultural and Animal Husbandry Sciences, Lasa 850032, China; zwangzhuang@163.com (Z.-Z.W.); qlxlsylzfyzx@163.com (W.-L.W.); lkk9992022@163.com (K.-K.L.); 2Key Laboratory of Fishery and Germplasm Resources Utilization of Xizang Autonomous Region, Lasa 850032, China; 3College of Animal Science and Technology, Henan Agricultural University, Zhengzhou 450046, China; 4Center for Research on Breeding and Utilization Techniques of Indigenous Fish Species in Xizang, Lasa 850032, China

**Keywords:** aquaculture, growth properties, gut microbiota, water environmental factors, Qinghai–Tibet Plateau

## Abstract

The gut microbiota plays a pivotal role in upholding intestinal health, fostering intestinal development, fortifying organisms against pathogen intrusion, regulating nutrient absorption, and managing the body’s lipid metabolism. However, the influence of different cultivation modes on the growth indices and intestinal microbes of *Salmo trutta fario* remains underexplored. In this study, we employed high-throughput sequencing and bioinformatics techniques to scrutinize the intestinal microbiota in three farming modes: traditional pond aquaculture (TPA), recirculating aquaculture (RA), and flow-through aquaculture (FTA). We aimed to assess the impact of different farming methods on the water environment and *Salmo trutta fario*’s growth performance. Our findings revealed that the final weight and weight gain rate in the FTA model surpassed those in the other two. Substantial disparities were observed in the composition, relative abundance, and diversity of *Salmo trutta fario* gut microbiota under different aquaculture modes. Notably, the dominant genera of *Salmo trutta fario* gut microbiota varied across farming modes: for instance, in the FTA model, the most prevalent genera were *SC-I-84* (7.34%), *Subgroup_6* (9.93%), and *UTCFX1* (6.71%), while, under RA farming, they were *Bacteroidetes_vadinHA17* (10.61%), *MBNT15* (7.09%), and *Anaeromyxoactor* (6.62%). In the TPA model, dominant genera in the gut microbiota included *Anaeromyxobacter* (8.72%), *Bacteroidetes_vadinHA17* (8.30%), and *Geobacter* (12.54%). From a comparative standpoint, the genus-level composition of the gut microbiota in the RA and TPA models exhibited relative similarity. The gut microbiota in the FTA model showcased the most intricate functional diversity, while TPA farming displayed a more intricate interaction pattern with the gut microbiota. Transparency, pH, dissolved oxygen, conductivity, total dissolved solids, and temperature emerged as pivotal factors influencing *Salmo trutta fario* gut microbiota under diverse farming conditions. These research findings offer valuable scientific insights for fostering healthy aquaculture practices and disease prevention and control measures for *Salmo trutta fario*, holding substantial significance for the sustainable development of the cold-water fish industry in the Qinghai–Tibet Plateau.

## 1. Introduction

*Salmo trutta fario*, a cold-water fish of the Salmoniformes Salmonidae family, also known as brown trout, is an imported species [1] native to Europe, northern Asia, and West Asia [2]. In China, it is only found in certain waters of Yadong County and Shigatse City, in the Xizang Autonomous Region [3]. Possessing a certain commercial value, it represents the most promising cold-water fish species in Xizang for industrialization [4], consequently facing overfishing. In 1992, it was classified as a second-tier key protected aquatic animal in this region. In recent years, both domestic and international scholars have predominantly focused on *Salmo trutta fario* in terms of the impact of organic matter and the role of physical factors in its healthy breeding ([5,6,7,8,9,10,11] and [12,13,14,15,16], respectively), the influence of different breeding methods [17,18], and its basic biology and disease prevention and control [19,20,21,22].

Fish gut microbiota plays a pivotal role in nutrient supply, metabolic balance, and immune defense, displaying a richer diversity to cope with the ever-changing ecological environment and food sources [23]. Fish harbor various gut microbiota types, including protozoa, fungi, viruses, and bacteria, with the latter prevailing as the dominant one in salmon intestines. Ongoing research, driven by technological advancements, continuously enhances our understanding of their origins, compositions, and functions and, both domestically and internationally, classifies them into two main categories: uncultured and cultured. This includes studies employing DNA sequencing technology and bioinformatics methods to scrutinize the characteristics of fish gut microbiota communities [24,25] and delve into their multifaceted functions, encompassing nutritional roles [26], immune modulation [27], and the interplay between them and aquatic environments [28,29]. Studies have unveiled that fish inhabit complex and fluctuating ecological niches, where alterations in habitat, temperature, feed, and intestinal structure can all prompt shifts in the diversity of their gut microbiota [30]. It has also been observed that maintaining a dynamic gut microbiota equilibrium profoundly influences the growth and development of fish and that a stable gut microbiota aids in the efficient digestion and absorption of nutrients, regulates the fish’s immune system, and preserves their overall health [31]. Nonetheless, investigations into the impacts of diverse aquaculture practices on *Salmo trutta fario*’s gut microbiota remain elusive.

This study represents the first comprehensive exploration into the impact of three distinct aquaculture methods—traditional pond aquaculture (TPA), recirculation aquaculture (RA), and flow-through aquaculture (FTA)—on the growth parameters, water quality indicators, and gut microbiota of *Salmo trutta fario*. By employing a biostatistical analysis, we scrutinized the differences in the growth metrics and aquatic environmental factors among *Salmo trutta fario* under varied cultivation modes. Utilizing high-throughput sequencing coupled with advanced bioinformatics, we delved into gut microbiota species composition, diversity, interspecies dynamics, and correlation with water environmental factors across different *Salmo trutta fario* aquaculture modes. This study holds significant implications for fostering *Salmo trutta fario* sustainable aquaculture and conservation, enhancing both economic and ecological outcomes, and laying the groundwork for exploring novel, safe, and effective approaches to aquatic disease prevention and control. It stands poised to propel the sustainable progression of the high-altitude cold-water fish aquaculture industry.

## 2. Materials and Methods

### 2.1. Aquaculture Farming Setup

This study received approval from the Fisheries Science Research Institute, at the Tibet Academy of Agricultural and Animal Husbandry Sciences. It was conducted in the Yarlung Tsangpo River Fish Breeding Base (29.638593° N, 91.030090° E), where the terrain slopes from east to west, characterized by a plateau temperate semi-arid monsoon climate, with an annual sunshine duration exceeding 3000 h. The site’s altitude is 3650 m, with an atmospheric pressure of 652.0 hPa. Fifteen healthy *Salmo trutta fario* of similar weights (Table 1) were sourced from the institute and randomly assigned to three groups (*n* = 5 each): traditional pond aquaculture (TPA), recirculation aquaculture (RA), and flow-through aquaculture (FTA).

The recirculation aquaculture (RA) model employed rectangular 6 m × 2.5 m × 2.0 m tanks, maintaining a water level of approximately 1 m, accommodating a total water volume of 12 m^3^. Within these tanks, three cages (each measuring 1.8 m × 2.5 m × 2.0 m) were installed, facilitating a water exchange rate ranging from 0 to 0.96 m^3^ per cycle, with a flow velocity from 0 to 0.144 m^3^/h. The flow-through aquaculture (FTA) model utilized cylindrical glass tanks (r = 0.6 m and h = 1 m), maintaining a water level of 0.8 m and featuring a flow rate ranging from 0 to 4.5 m^3^/h. The traditional pond aquaculture (TPA) model occupied an area of 5 m × 20 m. Groundwater drawn from the same source served as the water for all three cultivation methods, being pumped into reservoirs for aeration and sedimentation. In recurrent aquaculture (RA), water is directly pumped from the source, whereas, in flow-through aquaculture (FTA), it flows into the cultivation tanks from higher to lower elevations. Fish were fed twice daily, at 10:00 AM and 5:00 PM, with premium feed pellets containing crude protein ≥ 42%, crude fat ≥ 22%, carbohydrates ≥ 19%, crude fiber ≥ 2%, crude ash ≤ 6%, moisture content ≤ 8%, and total phosphorus ≥ 1%. The pre-trial phase spanned 7 days, followed by a 180-day experimental period.

### 2.2. Measurement of Growth Parameters and Aquaculture Water Environmental Factors

At the outset of this experiment, we measured the initial body weight using a digital precision balance accurate to 0.01 g, and after 180 days of cultivation, we assessed the final one using the same method. Upon the conclusion of this experiment, we computed the salmon survival rate (SR) and total weight gain rate (WGR).
GR (%) = final number of survivors/total number of deaths %
TWG (g) = initial body weight (g)/final body weight (g) %

To evaluate the post-experiment water environmental factors across the three models, transparency (Secchi disc, Jingcheng, Wuhan, China), pH (DR portable water quality detector, Hach, Loveland, CO, USA), dissolved oxygen (DR portable water quality detector, Hach, Loveland, CO, USA), conductivity (conductivity meter, SMART SENSOR, Shanghai, China), total dissolved solids (HI98302 pen type TDS tester, Hanna, Rome, Italy), and water temperature (electronic thermometer, OMRON, Kyoto, Japan) were directly assessed three times in the water under study, following the operational guidelines provided for each measuring instrument, and the results were averaged.

### 2.3. Collection and DNA Extraction of Intestinal Samples

After anesthetizing *Salmo trutta fario* from different farming modes using MS-222 (130 mg/L), intestinal contents were dissected, collected from 1 cm behind the rectum, and transferred into enzyme-free sterile EP tubes. These samples were promptly placed in dry ice and transported to the laboratory for storage at −80 °C. Meanwhile, water samples were collected using a vacuum pump (2 L) and filtered through a 0.22-micron mixed acetic acid-digested cellulose filter membrane. The filtered water was then transferred into enzyme-free sterile EP tubes and stored at −80 °C. We extracted total genomic DNA samples using the OMEGA DNA Kit (M5635-02) (Omega Bio-Tek, Norcross, GA, USA), following the manufacturer’s instructions, and stored them at −20 °C for further analysis. The quantity and quality of the DNA were assessed using a NanoDrop NC2000 spectrophotometer (Thermo Fisher Scientific, Waltham, MA, USA) and agarose gel electrophoresis, respectively.

### 2.4. Amplification and High-Throughput Sequencing of Intestinal Microbiota 16S rRNA

Sequencing was conducted by Personal Biotechnology Co. Ltd. (Shanghai, China). PCR amplification targeted the V3–V4 region of the bacterial 16S rRNA gene, utilizing forward primer 338F (5′-ACTCCTACGGGAGGCAGCA-3′) and reverse primer 806R (5′-GGACTACHVGGGTWTCTAAT-3′). Sample-specific 7 bp barcodes were incorporated into the primers for multiplex sequencing. The PCR reaction mixture comprised 5 μL of 5× buffer, 0.25 μL of Fast Pfu DNA Polymerase (5 U/μL), 2 μL of dNTPs (2.5 mM), 1 μL of each forward (10 µM) and reverse primer (10 µM), 1 μL of DNA template, and 14.75 μL of ddH_2_O. The thermal cycling conditions were as follows: initial denaturation at 98 °C for 5 min, followed by 25 cycles of denaturation at 98 °C for 30 s, annealing at 53 °C for 30 s, and extension at 72 °C for 45 s, with a final extension step at 72 °C for 5 min. The PCR amplicons were purified using Vazyme VAHTSTM DNA Clean Beads (Vazyme, Nanjing, China) and quantified with the Quant-iT PicoGreen dsDNA Assay Kit (Invitrogen, Carlsbad, CA, USA). Subsequently, the amplicons were pooled in equal proportions and subjected to paired-end 2250 bp sequencing on the Illumina NovaSeq platform using the NovaSeq 6000 SP (Illumina, San Diego, CA, USA.) Reagent Kit (500 cycles).

### 2.5. Statistical Analysis

Statistical analyses were conducted in R-4.2.1 (https://cran.r-project.org/ (accessed on 21 June 2022)). ASV stands for Amplicon Sequence Variant, and ASV data analysis was performed using the dada2 package in R (https://benjjneb.github.io/dada2/ (accessed on 3 April 2024)). The one-way ANOVA method was employed to analyze both the growth parameters of fish and the physicochemical factors of water, utilizing the SPSS 21.0 software. The alpha diversity indices (including Observed, Shannon–Wiener diversity index, Chao1 index, ACE, Simpson dominance index, and Fisher) were calculated using the “vegan” package in R [32]. Principal coordinate analysis (PCoA) was performed based on the Bray–Curtis distance using the “vegan” package in R. Before depicting the heatmap, we standardized the abundance data of the microbial communities (using the scale the package in R, https://blog.csdn.net/ByteNinja/article/details/132518709 (accessed on 3 April 2024)). The intestinal microbiota’s co-occurrence patterns were constructed based on Spearman’s rank correlation coefficients. The co-occurrence network was visualized in Gephi (version 0.9.2) [33]. We utilized the Linear Discriminant Analysis Effect Size (LEfSe) to compare the key differential groups and functional predictions of *Salmo trutta fario* gut bacteria across the three aquaculture farming modes (using the microeco package in R, https://github.com/ChiLiubio/microeco (accessed on 3 April 2024)). Mantel tests were used to determine correlations between environmental variables and selected characteristics of the intestinal microbiota’s composition in the “linkET” package in R (linkET: Everything is Linkable. R package version 3.0.3. https://github.com/Hy4m/linkET (accessed on 22 April 2024)). The FAPROTAX database is a collection of prokaryotes’ traits and functions based on the known research results published in books and the literature.

## 3. Results

### 3.1. Varied Growth Performance Metrics among Salmo trutta fario Populations under Various Aquaculture Modes

The growth phenotype data for the three *Salmo trutta fario* groups are presented in Table 1. No significant differences were observed in the initial culture density or body weight among the three groups (*p* > 0.05). However, the final body weight in the RA group was significantly lower than that in the TPA group (*p* < 0.01), while the FTA group’s was the highest of the three (*p* < 0.01). These results indicate that the specific growth rate (GR) was significantly higher in the RA and FTA groups compared to the TPA group, with no significant difference observed between the RA and FTA groups. Furthermore, significant differences in the weight gain rate (WGR) were noted among the three groups: the RA group exhibited a lower WGR compared to the TPA group (*p* < 0.01), whereas the FTA group’s WGR was the highest of the three (*p* < 0.01).

### 3.2. Highly Variable Diversity in Salmo trutta fario Intestinal Microbiota Communities across Various Aquaculture Modes

Using 16S sequencing at the ASV level, the FTA group exhibited the highest number of ASVs, with 4716 identified, followed by the RA group, with 3677 ASVs, and the TPA group, with 3619 ASVs. The sequencing coverage of all 15 samples exceeded 97%, indicating the comprehensive detection of intestinal microorganisms in each *Salmo trutta fario* specimen.

*Salmo trutta* gut microbiota’s alpha diversity varied across different farming modes (Figure 1A, Appendix A): the observed ASV counts ranged from 668 to 1629; the Chao1 index varied from 687.41 to 1955.55; the ACE index ranged from 688.66 to 1952.20; the fragrant aroma diversity index varied from 6.03 to 6.99; Simpson’s index fluctuated between 0.994 and 0.999; and Fisher’s index value ranged from 183.17 to 676.41. Under the FTA mode, the average alpha diversity indices were consistently higher compared to the other two models, with TPA showing relatively higher indices than RA, even though the difference is not statistically significant. The PCoA analysis based on the Bray–Curtis distance indicated that PCoA1 and PCoA2 explained 24.6% and 20.9% of gut microbiota variation, respectively. Significant differences in *Salmo trutta fario* gut microbiota were observed with changes in the aquaculture mode (*p* < 0.05) (Figure 1B).

### 3.3. Variations in Salmo trutta fario Intestinal Microbiota Community across Various Aquaculture Modes

Variability in the composition and relative abundance of *Salmo trutta fario* gut microbiota was pronounced as the farming methods shifted (Figure 2): RA exhibited 3085 unique ASVs and shared 158 ASVs with TPA, 267 ASVs with FTA, and 147 ASVs with both; TPA featured 3049 unique ASVs and shared 265 ASVs with FTA; and FTA harbored 4037 unique ASVs (Figure 2A).

Additionally, we analyzed the phylum-level relative abundance of *Salmo trutta fario* gut microbiota across the various farming modes (Figure 2B, Appendix A): in FTA, the predominant phyla were Proteobacteria (49.29%), Acidobacteria (13.88%), and Chloroflexi (10.59%); in RA, they were Proteobacteria (39.01%), Chloroflexi (19.09%), and Acidobacteria (14.52%); meanwhile, in TPA, the primary phyla were Proteobacteria (41.86%), Chloroflexi (23.12%), and Acidobacteria (14.08%). At the genus level, we chose the top ten species based on their relative abundance rankings (Figure 2C, Appendix A): within FTA, the most abundant genera were *SC-I-84* (7.34%), *Subgroup_6* (9.93%), and *UTCFX1* (6.71%); under RA farming, they were *Bacteroidetes_vadinHA17* (10.61%), MBNT15 (7.09%), and *Anaeromyxobacter* (6.62%); and, in TPA, they were *Anaeromyxobacter* (8.72%), *Bacteroidetes_vadinHA17* (8.30%), and *Geobacter* (12.54%).

At the genus level, a clustered heatmap was generated to depict the similarities and differences in the composition of the top 20 species with the highest relative abundance of gut microbiota in *Salmo trutta fario* across three farming modes, utilizing color variations and similarity metrics (Figure 2D): RA and TPA cluster closely together, indicating a relatively similar composition, subsequently clustering together with FTA, which showed a comparatively distinct composition.

### 3.4. Distribution and Functional Prediction of Core Microorganisms in Salmo trutta fario Intestinal Tract across Various Aquaculture Modes

To further elucidate the core microbial community within *Salmo trutta fario* intestinal microbiota across the various aquaculture modes, we employed an LEfSe analysis to compare the abundance changes, setting *p* < 0.05 and an LDA value > 4 as the criteria for significant differences.

g_Bacteroidetes vadinHA17 showed significant enrichment in *Salmo trutta fario*‘s intestinal tract under RA farming, whereas g_Geobacter, g_Sva0485, g_RBG-13-54-9, and g_Anaeromyxobacter were notably enriched under the TPA model and g_Rokubacteriales, g_MND1, and g_Defluviicoccus under FTA farming (Figure 3A, Appendix A).

The functional gene prediction of *Salmo trutta fario* gut microbiota under the three farming modes revealed that the predominant microbial functions in the gut remained consistent, encompassing aerobic chemoheterotrophy, iron respiration, chemoheterotrophy, nitrification, anaerobic chemoheterotrophy, fermentation, and aerobic ammonia oxidation. Notably, functional diversity peaked under FTA farming (Figure 3B, Appendix A).

### 3.5. Co-Occurrence Network Analysis of Salmo trutta fario Intestinal Microbiota Community across Diverse Aquaculture Modes

We conducted a co-occurrence network analysis to delve into potential relationships among the *Salmo trutta fario* gut microbiota communities across the three farming methods. The modularization coefficients of all six co-occurrence networks surpassed 0.4, signifying notable modularized structures. Distinct network metrics were noted across the three farming methods, primarily varying in node and edge numbers, indicating dynamic fluctuations (Figure 4A). In TPA farming, the co-occurrence network showed the highest total number of edges and average degree (the average number of connections per node in the network), suggesting that interactions within *Salmo trutta fario*’s gut microbiota community are more intricate under this mode compared to the other two.

To delve deeper into these co-occurrence relationships, we examined the interactions among different taxa. Our analysis revealed distinct interaction patterns for each farming mode, with the top four co-occurring taxa as follows: RA farming, Proteobacteria (29.71%) > Chloroflexi (22.06%) > Actinobacteria (16.47%) > Acidobacteria (13.53%); TPA farming, Proteobacteria (35.02%) > Chloroflexi (19.82%) > Acidobacteria (12.44%) > Bacteroidetes (8.06%); and FTA farming, Proteobacteria (44.95%) > Acidobacteria (15.66%) > Chloroflexi (10.35%) > Actinobacteria (6.82%) (Figure 4B). Nodes belonging to the Proteobacteria taxa demonstrated a higher proportion of interactions with species from other taxa, indicating that Proteobacteria played a significant role in shaping the network structure across all three farming modes.

### 3.6. Different Aquaculture Modes Impact Salmo trutta fario Gut Microbiota Community Characteristics through Aquatic Environmental Factors

The mean values of aquaculture water environmental factors are compared across the different farming modes in Table 2 (Appendix A), with transparency, pH, dissolved oxygen, conductivity, total dissolved solids, and temperature showing varying significant differences (*p* < 0.05): transparency, pH, and temperature are notably higher in FTA; dissolved oxygen and conductivity are significantly higher in RA (*p* < 0.05); and the total dissolved solids are significantly elevated in TPA (*p* < 0.05).

To investigate the pivotal factors shaping *Salmo trutta fario*’s gut microbiota community, we utilized Mantel’s analysis to assess the impact of various aquaculture water environmental factors on the gut microbiota community (Figure 5). In RA farming, transparency, dissolved oxygen, conductivity, and temperature emerged as key factors influencing *Salmo trutta fario* gut microbiota and were positively correlated with its alpha diversity index. In the TPA model, dissolved oxygen and total dissolved solids were the primary factors positively affecting *Salmo trutta fario* gut microbiota, while, within FTA, these were transparency, pH, total dissolved solids, and temperature.

## 4. Discussion

### 4.1. Differences in Salmo trutta fario Gut Microbiota Characteristics and Distribution Patterns across Various Aquaculture Modes

Fish harbor diverse and abundant microbial communities within their intestines that contribute to the development and health of the intestines by establishing a complex and stable dynamic balance with the intestinal tissue and contents. The intestinal microbiota plays a crucial role in improving and maintaining fish intestinal environment, facilitating nutrient digestion, boosting fish immunity, and sustaining overall body health [34]. Studies have identified Proteobacteria, Clostridia, Firmicutes, Bacteroidetes, Actinobacteria, Clostridia, Bacilli, and Verrucomycota as the main phyla in fish gut [35,36]. Among these, Firmicutes, Proteobacteria, and Bacteroidetes are consistently found across various fish intestines, constituting the core microbial communities that effectively modulate the diversity and structure of fish gut microbiota.

In our investigation, similar findings were obtained. At the phylum level, *Salmo trutta fario* gut microbiota under three different farming modes primarily consisted of Proteobacteria, Acidobacteria, and Chloroflexi, aligning with previous results indicating that the dominant gut microbe group in Nayak fish comprises bacteria [25] and with Roeselers G, who identified similar core microbial communities in zebrafish (Daniorerio) gut microbiota under diverse environmental conditions [37]. The findings of the current study contrast with those of Li et al. on the predominant intestinal microorganism categories in grass carp [38]. However, in our study, differences in species composition and relative abundance were observed at the genus level among *Salmo trutta fario* gut microbiota core communities under varied aquaculture modes, including differences in α and β diversity, with the analysis chart indicating distinct dispersion of the three sample groups, signifying significant differences in composition. The hierarchical clustering analysis yielded similar results. These variations may stem from the intricate and ever-changing ecological settings that fish inhabit, where alterations in habitat, temperature, feed, and intestinal structure can influence the diversity in their gut microbiota [30].

Research findings indicate that increased bacterial interactions and the sharing of niches contribute to the expansion and complexity of microbial networks, highlighting fundamental distinctions among various microbial samples [39]. Our study revealed divergent co-occurrence networks in *Salmo trutta fario* gut microbiota under three distinct aquaculture modes, suggesting that different farming practices may influence its stability. Notably, FTA exhibited optimal connectivity, transferability, clustering probability, and compactness among microbial community nodes, indicating superior stability compared to the other models. Conversely, TPA demonstrated a higher total number of edges and average degree, reflecting a more intricate network of interactions within the gut microbiota community. Within a network module, a highly connected group is termed a module center point, whose reduction in loss can disrupt both the module’s and network’s integrity, potentially leading to an imbalance in the microbial community and impacting host health [40]. In summary, aquaculture practices significantly influence *Salmo trutta fario* gut microbiota species composition, community structure, diversity, and distribution patterns.

Nevertheless, this study might have been constrained by a small sample size or limited sampling sources (*n* = 5), potentially restricting the generalizability and robustness of the findings. Furthermore, ethical approval and adherence to regulatory standards are typically obligatory for animal experiments, imposing additional costs and time constraints, thereby limiting the scope of sample collection. Finally, considering the experimental novelty of our study, we are committed to refining this research in future endeavors.

### 4.2. Differences in Salmo trutta fario Growth Indicators and Aquaculture Water Environment across Various Aquaculture Modes

Across various aquaculture modes, fish are subject to a range of factors, such as water quality and feed, that can lead to fluctuations in their growth performance and metrics. Our investigation unveiled notable disparities in the final weight and survival and total weight gain rates of *Salmo trutta fario* under different aquaculture modes, with consistently superior metrics being observed in the FTA group compared to the other two. Past studies have underscored the efficacy of RA in mitigating significant challenges linked to cage farming, encompassing eutrophication and the potential spread of invasive species, diseases, and antibiotic resistance [41]. Additionally, research suggests that alterations in natural water flow can influence a water body’s temperature and dissolved oxygen levels, thereby impacting salmon development [42]. It is worth noting that the optimal growth temperature for *salmon* typically hovers around 12 °C, with this ideal temperature decreasing as fish size increases, ultimately influenced by the water’s oxygen content [43,44,45]. Our findings revealed that both the FTA and RA groups exhibited elevated levels of dissolved oxygen in the water compared to the TPA group, with FTA also demonstrating a higher water temperature. These observations further elucidate the disparities in the *Salmo trutta fario* growth phenotypes observed across different cultivation modes in our experiments.

### 4.3. Water Environmental Factors: Key Influences on Salmo trutta fario Gut Microbiota across Various Aquaculture Modes

The water environment offers a rich and intricate habitat for fish gut microbiota, whose diversity is intricately linked to factors such as microorganisms, salinity, water temperature, dissolved oxygen, and water depth [28]. Fluctuations in these water environmental factors prompt corresponding shifts in the gut microbiota, consequently impacting fish immune and metabolic systems. Notably, significant alterations in the gut microbiota triggered by changes in water environmental factors can serve as biomarkers for detecting shifts in the aquatic environment. Studies have established a close relationship between a water body’s salinity and temperature and fish gut and water microbiota [23,46,47]. Moderate pH levels, salinity, and other physicochemical factors within the water environment foster the growth of intestinal probiotics, exerting beneficial effects on fish [36]. Consistent with prior research, our study revealed a positive correlation between the aquaculture water environment and *Salmo trutta fario* gut microbiota across three distinct aquaculture modes; transparency, pH, dissolved oxygen, conductivity, total dissolved solids, and temperature emerged as the key influencing factors. It is noteworthy that different fish species exhibit varying adaptability and tolerance to water environmental changes, consequently leading to alterations in their gut microbial niche. While this study explored the influence of water quality parameters on *Salmo trutta fario* intestinal microbiome, it is important to note that other environmental variables whose comprehensive consideration was lacking in our study, such as breeding density and feed composition, could also affect the outcomes. We remain committed to refining this experiment in subsequent phases to address these factors.

## 5. Conclusions

This study represented the first application of ecological, bioinformatic, and high-throughput sequencing methodologies to explore the impact of various cultivation modes on *Salmo trutta fario* growth performance and intestinal microbiota. Our findings underscored the FTA model’s propensity for fostering weight gain in *Salmo trutta fario*. Noteworthy differences were observed in the composition, relative abundance, community structure, diversity, distribution pattern, core community functions, and interspecific relationships of *Salmo trutta fario* gut microbiota across the distinct aquaculture modes. The correlation between the environmental factors in the three farming models and *Salmo trutta fario* gut microbiota suggested the existence of a longstanding adaptive mechanism between microbial communities and their ecological niche.

## Figures and Tables

**Figure 1 microorganisms-12-01082-f001:**
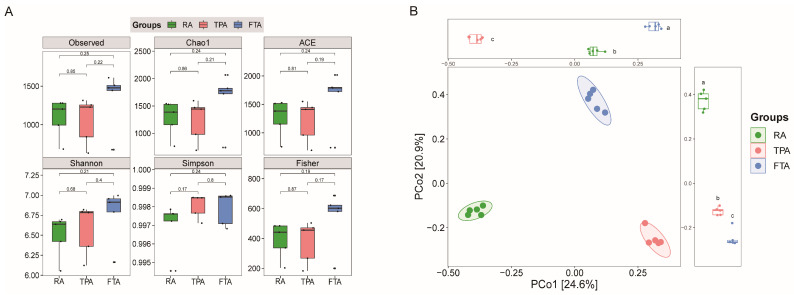
Diversity in *Salmo trutta fario* gut microbiota across various aquaculture modes: (**A**) alpha diversity and (**B**) beta diversity. The lowercase letters in the figure indicate distinct levels of significance (*p* < 0.05).

**Figure 2 microorganisms-12-01082-f002:**
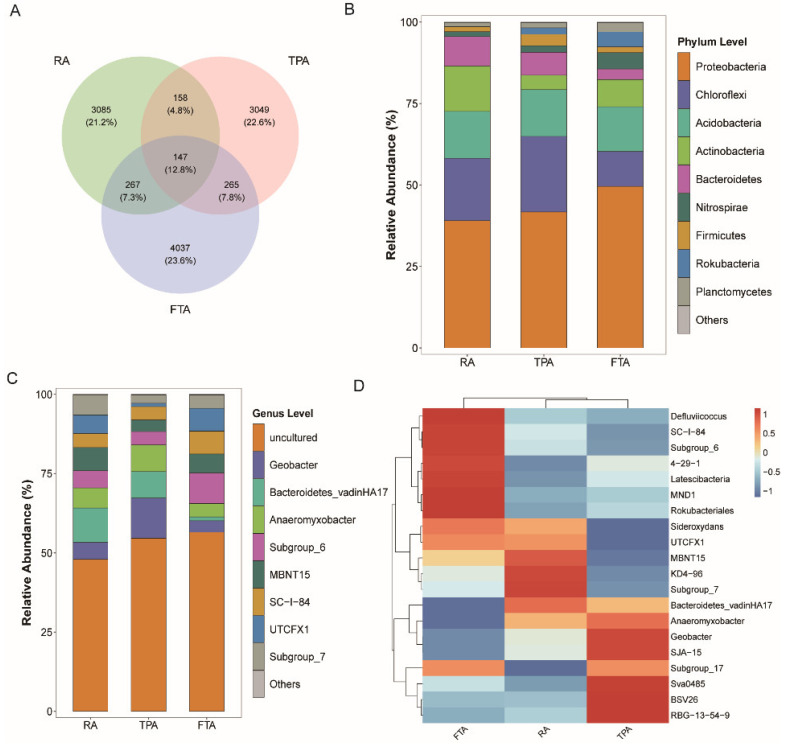
Variations in species distribution and relative abundance of gut microbiota in *Salmo trutta fario* across different farming modes: (**A**) Venn diagram; (**B**) phylum-level relative abundance; (**C**) genus-level relative abundance; and (**D**) heatmap depicting the relative abundance of major genus-level groups, based on gut microbiome species abundance data.

**Figure 3 microorganisms-12-01082-f003:**
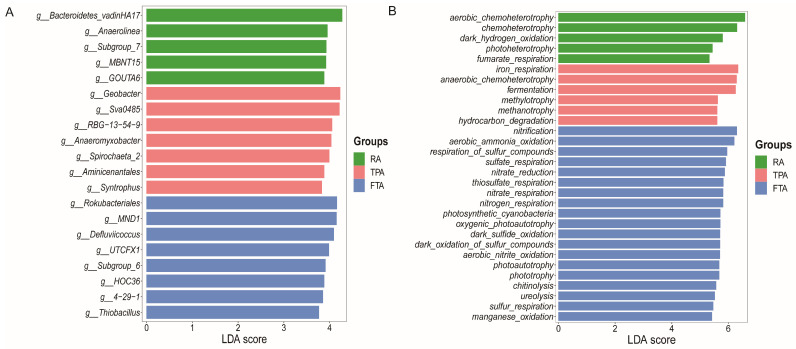
Distribution (**A**) and functional prediction (**B**) of the primary gut microbiota in *Salmo trutta fario* across various farming modes.

**Figure 4 microorganisms-12-01082-f004:**
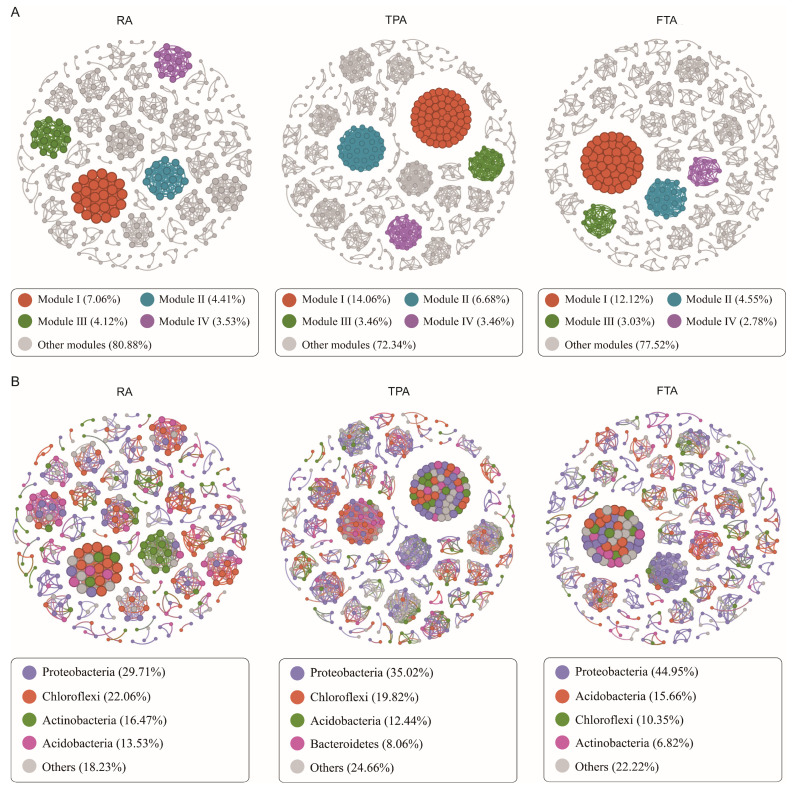
Co-occurrence network analysis of *Salmo trutta fario* gut microbiota across three farming modes: (**A**) co-occurrence networks of gut microbiota and (**B**) co-occurrence relationships within the gut microbial communities.

**Figure 5 microorganisms-12-01082-f005:**
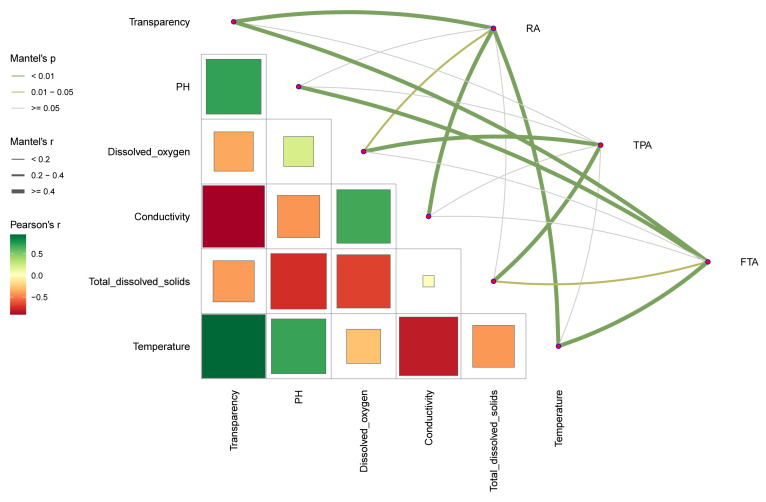
Environmental drivers of *Salmo trutta fario* intestinal microbiota community across diverse aquaculture modes. This primarily highlights the correlation between environmental parameters and the diversity index of the gut microbial community in *Salmo trutta fario* under different farming conditions.

**Table 1 microorganisms-12-01082-t001:** Salmo trutta fario phenotypic growth data under various aquaculture modes.

Aquaculture Modes	Density (kg/m^3^)	Initial Body Weight (g)	Final Body Weight (g)	GR%	WGR%
TPA	7.83 ± 0.06 a	100.35 ± 2.89 a	363.00 ± 4.56 b	85.46 ± 1.23 b	262.34 ± 7.45 b
RA	7.81 ± 0.02 a	103.04 ± 2.85 a	315.29 ± 3.26 c	97.08 ± 1.91 a	206.13 ± 5.59 c
FTA	7.85 ± 0.03 a	102.67 ± 3.08 a	423.72 ± 3.34 a	99.17 ± 0.72 a	314.10 ± 9.46 a

Note: Different letters indicate significance levels (ANOVA Duncan’s multiple comparisons, significance level 0.05).

**Table 2 microorganisms-12-01082-t002:** Aquatic environmental factors across different farming modes.

Aquaculture Modes	Transparency (cm)	pH	Dissolved Oxygen (mg/L)	Conductivity (μS/cm)	Total Dissolved Solids (mg/L)	Temperature (°C)
TPA	0.32 ± 0.02 c	7.60 ± 0.12 b	6.67 ± 0.19 b	306.80 ± 4.87 b	211.67 ± 6.69 a	9.30 ± 0.15 c
RA	0.15 ± 0.01 b	7.47 ± 0.09 b	7.77 ± 0.09 a	362.20 ± 5.63 a	195.33 ± 3.28 a	7.07 ± 0.15 b
FTA	0.48 ± 0.02 a	8.20 ± 0.06 a	7.40 ± 0.06 a	296.33 ± 3.71 b	174.67 ± 3.48 b	11.57 ± 0.41 a

Note: Different letters indicate significance levels (ANOVA Duncan’s multiple comparisons, significance level 0.05).

## Data Availability

The original contributions presented in the study are included in the article/Appendix A, further inquiries can be directed to the corresponding author.

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
