# Peer review of "Effects of Different Farming Modes on Salmo trutta fario Growth and Intestinal Microbial Community"

_microorganisms, 2024, doi:10.3390/microorganisms12061082_

Round 1

Reviewer 1 Report

Comments and Suggestions for Authors

The current study investigated the impact of three 75 distinct aquaculture methods – traditional pond aquaculture (TPA), recurrent aquaculture 76 (RA), and flow-through aquaculture (FTA) – on the growth parameters, water quality in-77 dicators, and gut microbiota of Salmo trutta fario.

1.      Line 159-160: what packages were used to perform LEfSe? And please cite the package.

2.      Line 183: What package(s) were used to process ASV? It was not mentioned in the method. Also define ASV in the first instance.

3.      Line 192-193: Authors claimed, “Under the 192 FTA aquaculture mode, alpha diversity indices are consistently higher compared to the 193 other two aquaculture models”, yet in Figure 1, none of the comparison has a significant p-value in any of the 6 metrics.

4.      Figure 2C: is it uncultured or unclassified? Some bacteria could be cultured but unclassified due to poor sequence quality.

5.      Figure 2C, D: Why are there more genera in 2D than in 2C? Both are the sample genus level. Also explain what metrics were used to plot the heatmap in the figure legends.

6.      Line 251: How are functional genes predicted? It was not mentioned anywhere in the method.

7.      Line 305-312: “Transparency, Dissolved Oxygen, Conductivity, and Temperature emerged as key factors 306 influencing the gut microbiota of Salmo trutta fario, exhibiting a positive correlation.” Those environmental factors positively correlated with what metrics in the microbiota?

8.      Line 325-327: Neither citation (33 nor 34) supports the claim about fish gut.

9.      Is there no fish gut study other than citation 35? Please discuss more related studies.

10.  In the discussion, the authors should also discuss the limitations of the study. For example, n=5 is quite limited.

Minor:

11.  Line 45: Should be imported species “in China”.

12.  Line 155: Provide citation for vegan package.

13.  Line 159:  Please provide citation for Gephi software. Also missing a space in between.

14.  Line 163: Provide citation to the LinkET package.

15.  Line 305: Mantenl is a typo.

Author Response

Reviewer #1

The current study investigated the impact of three 75 distinct aquaculture methods – traditional pond aquaculture (TPA), recurrent aquaculture 76 (RA), and flow-through aquaculture (FTA) – on the growth parameters, water quality in-77 dicators, and gut microbiota of Salmo trutta fario.

Response: We greatly appreciate the reviewers' valuable comments and suggestions on our manuscript. In response, we have conducted comprehensive revisions to align with your requirements and recommendations. All modifications have been clearly highlighted for your convenience.

  1. Line 159-160: what packages were used to perform LEfSe? And please cite the package.

Response: The manuscript has been revised according to the reviewer's comments. We supplemented the manuscript with the microeco package in R(https://github.com/ChiLiubio/microeco) used in performing LEfSe.

  1. Line 183: What package(s) were used to process ASV? It was not mentioned in the method. Also define ASV in the first instance.

Response: The manuscript has been revised according to the reviewer's comments. The ASV data were processed using the dada2 package in R, as detailed in the Methods section. ASV stands for Amplicon Sequence Variant, and its definition will be provided in the initial instance.

  1. Line 192-193: Authors claimed, “Under the 192 FTA aquaculture mode, alpha diversity indices are consistently higher compared to the 193 other two aquaculture models”, yet in Figure 1, none of the comparison has a significant p-value in any of the 6 metrics.

Response: The manuscript has been revised according to the reviewer's comments. While there were no statistically significant differences observed in the six measures of Alpha diversity (P>0.05), the average Alpha diversity index was higher under the FTA aquaculture model compared to the other two aquaculture models. In the manuscript, we have revised to reflect that the average alpha diversity index of the FTA aquaculture model was higher than that of the other two aquaculture models.

  1. Figure 2C: is it uncultured or unclassified? Some bacteria could be cultured but unclassified due to poor sequence quality.

Response: In Figure 2C, the term "uncultured" is used to denote bacteria that were not successfully cultured in the laboratory setting.

  1. Figure 2C, D: Why are there more genera in 2D than in 2C? Both are the sample genus level. Also explain what metrics were used to plot the heatmap in the figure legends.

Response: Thank you to the reviewer for the valuable suggestions. We have analyzed Figure 2 of the manuscript and revised the content accordingly. Although both Figures 2C and 2D are analyzed at the genus level, we selected different species relative abundance rankings based on the distinct functionalities presented by the two figures. We used the species abundance of the gut microbiome to create the heatmap and included explanations in the figure legend of the manuscript.

  1. Line 251: How are functional genes predicted? It was not mentioned anywhere in the method.

Response: We incorporated a method for predicting functions into the statistical analysis section of the Materials and Methods

  1. Line 305-312: “Transparency, Dissolved Oxygen, Conductivity, and Temperature emerged as key factors 306 influencing the gut microbiota of Salmo trutta fario, exhibiting a positive correlation.” Those environmental factors positively correlated with what metrics in the microbiota?

Response: Thank you to the reviewer for the valuable suggestions. We've adjusted the manuscript accordingly. It appears that there's a positive correlation between the alpha diversity index of gut microbiota and aquatic environmental factors.

  1. Line 325-327: Neither citation (33 nor 34) supports the claim about fish gut.

Response: Thank you to the reviewer for the valuable suggestions. We've reviewed citations 33 and 34 and updated them with the accurate references.

  1. Is there no fish gut study other than citation 35? Please discuss more related studies.

Response: Thank you to the reviewer for the valuable suggestions. We have included additional discussion in the manuscript.

  1. In the discussion, the authors should also discuss the limitations of the study. For example, n=5 is quite limited.

Response: Thank you to the reviewer for the valuable suggestions. We have incorporated a discussion regarding the limitations of this study into the manuscript. We outlined the limitations of this study, highlighting constraints such as sample size and the intricate nature of environmental factors impacting gut microbes.

Minor:

  1. Line 45: Should be imported species “in China”.

Response: Modifications have been implemented based on your review suggestions.

  1. Line 155: Provide citation for vegan package.

Response: References have been included in accordance with your review suggestions.

  1. Line 159: Please provide citation for Gephi software. Also missing a space in between.

Response: References have been included in accordance with your review suggestions.

  1. Line 163: Provide citation to the LinkET package.

Response: References have been included in accordance with your review suggestions.

  1. Line 305: Mantenl is a typo.

Response: Modifications have been implemented based on your review suggestions.

In addition, We edited the manuscript in English according to the links provided by the MDPI platform. English editorial number: english-edited-80845.

Reviewer 2 Report

Comments and Suggestions for Authors

The part of your MS about the investigation of the microbiome of the fish is well done. However, there are serious insufficiencies in materials and methods where the correct description of the conditions of fishkeeping (and the correct numbers) is practically missing. Some minor shortcomings and faults are marked in the attached file.

Comments on the Quality of English Language

A thorough overview by a native English speaker would really be needed.

Author Response

Reviewer #2

The part of your MS about the investigation of the microbiome of the fish is well done. However, there are serious insufficiencies in materials and methods where the correct description of the conditions of fishkeeping (and the correct numbers) is practically missing. Some minor shortcomings and faults are marked in the attached file.

Response: We greatly appreciate the reviewers' valuable comments and suggestions on our manuscript. In response, we have conducted comprehensive revisions to align with your requirements and recommendations. We have meticulously refined the experimental method, particularly focusing on the conditions of fish culture. All modifications have been clearly highlighted for your convenience.

A thorough overview by a native English speaker would really be needed.

Response: We edited the manuscript in English according to the links provided by the MDPI platform. English editorial number: english-edited-80845.

Comments on amendments to the annex to the review report:

  1. What does it mean?

Response: Thank you to the reviewer for the valuable suggestions. The ongoing research on fish gut microbiota, both domestically and internationally, classifies bacteria into two main categories: uncultured bacteria and cultured bacteria. Culturable bacteria refer to those bacteria that can be cultured and propagated in laboratory conditions using media such as agar. These bacteria can grow and form visible colonies under appropriate nutritional conditions, such as temperature, pH, and oxygen concentration. On the other hand, unculturable bacteria are a type of bacteria that cannot be cultured using traditional methods. These bacteria may require specialized environmental conditions or growth requirements that cannot be provided in the laboratory. The study and detection of unculturable bacteria often rely on modern biotechnological techniques such as gene sequencing and fluorescence microscopy. These methods allow for the direct analysis and study of bacteria without the need for culturing them.

  1. Recirculation

Response: Modifications have been implemented based on your review suggestions.

  1. Differencies

Response: Modifications have been implemented based on your review suggestions.

  1. From where the data of Table 1 came from?

Response: Thank you to the reviewer for the valuable suggestions. The initial weight of the fish can be found in Table 1.

  1. How fish were kept? How could you modeling the 3 technologies?No repetitions of the treatments (modes) were applied?

Response: Thank you to the reviewer for the valuable suggestions. We have added the corresponding information in the manuscript: The recirculation aquaculture (RA) model employed rectangular 6 m×2.5 m×2.0 m tanks, maintaining a water level of approximately 1 m, accommodating a total water volume of 12 m3. Within these tanks, three cages (each measuring 1.8 m×2.5 m×2.0 m) were installed, facilitating a water exchange rate ranging from 0 to 0.96 m3 per cycle, with a flow velocity from 0 to 0.144 m3/h. The flow-through aquaculture (FTA) model utilized cylindrical glass tanks (r=0.6 m and h=1 m), maintaining a water level of 0.8 m and fea-turing a flow rate ranging from 0 to 4.5 m3/h. The traditional pond aquaculture (TPA) model occupied an area of 5 m×20 m. Groundwater drawn from the same source served as the water for all three cultivation methods, being pumped into reservoirs for aeration and sedimentation. In recurrent aquaculture (RA), water is directly pumped from the source, whereas, in flow-through aquaculture (FTA), it flows into the cultivation tanks from higher to lower elevations. Fish were fed twice daily, at 10:00 AM and 5:00 PM, with premium feed pellets containing crude protein ≥42%, crude fat ≥22%, carbohydrates ≥19%, crude fiber ≥2%, crude ash ≤6%, moisture content ≤8%, and total phosphorus ≥1%. The pre-trial phase spanned 7 days, followed by a 180-day experimental period.

  1. Composition of the feed?

Response: “crude protein ≥42%, crude fat ≥22%, carbohydrates ≥19%, crude fiber ≥2%, crude ash ≤6%, moisture content ≤8%, and total phosphorus ≥1%.” These are the main components of feed.

  1. SR was named previously as survival.

Response: Thank you to the reviewer for the valuable suggestions. We have revised 'SR' to 'GR' in the manuscript.

  1. In what units?

Response: Thank you to the reviewer for the valuable suggestions. We have included the unit information for environmental factors in the manuscript.

  1. How can be explained these differences? How the FTA's water can be the least cold?

Response: In flow-through aquaculture (FTA), water sources flow through ponds and may be exposed to sunlight, resulting in higher water temperatures compared to the other two aquaculture models.

  1. First

Response: Modifications have been implemented based on your review suggestions.

Round 2

Reviewer 1 Report

Comments and Suggestions for Authors

The authors have successfully addressed most of my concerns. There are a few minor corrections that can be made to improve the quality of the manuscript.

Line 227-230: Please add “even though the difference was not statistically significant” to the sentence.

Follow up to Figure 2C question:  You did not culture those bacteria but simply match the sequence to the database. So I suggest changing “uncultured” to “unmatched” or “unclassified”.

Figure 2D: author mentioned that “they used the species abundance of the gut microbiome to create the heatmap”. The scale is from -1 to 1 and the abundance cannot be negative. The author must have used some sort of transformation or normalization method, please specify the method in the figure legend.

Follow up to Mentel test: Mantel tests are correlation tests that determine the correlation between two matrices (rather than two variables).  So, the positive correlation probably should be between the environmental parameter distance matrix and species abundance dissimilarity matrix created using a distance measure, in your case, Bray-curtis dissimilarity.

Author Response

Reviewer #1

The authors have successfully addressed most of my concerns. There are a few minor corrections that can be made to improve the quality of the manuscript.

Response: We sincerely appreciate you taking the time out of your busy schedule to review and provide detailed comments on our manuscript. Your professional opinions and constructive feedback are invaluable to us, not only in identifying shortcomings in our work but also in guiding the direction for future improvements in our research. We are deeply impressed by your rigorous attitude and academic spirit, and we have implemented your suggestions to enhance the quality and academic value of our manuscript. We also look forward to your continued guidance and support in future research and collaborations. Thank you once again for your time and effort. All modifications have been clearly highlighted for your convenience.

  1. Line 227-230: Please add “even though the difference was not statistically significant” to the sentence.

Response: Thank you to the reviewer for the valuable suggestions. Modifications have been implemented based on your review suggestions.

  1. Follow up to Figure 2C question: You did not culture those bacteria but simply match the sequence to the database. So I suggest changing “uncultured” to “unmatched” or “unclassified”.

Response: Thank you to the reviewer for the valuable suggestions. Modifications have been implemented based on your review suggestions. Changed "uncultured" to "unclassified" in Figure 2C.

  1. Figure 2D: author mentioned that “they used the species abundance of the gut microbiome to create the heatmap”. The scale is from -1 to 1 and the abundance cannot be negative. The author must have used some sort of transformation or normalization method, please specify the method in the figure legend.

Response: Thank you to the reviewer for the valuable suggestions. We have included additional explanations in the Statistical Analysis section of the manuscript. Before depicting the heatmap, we standardized the abundance data of the microbial communities (using the scale the package in R, https://blog.csdn.net/ByteNinja/article/details/132518709). scale is generic function whose default method centers and/or scales the columns of a numeric matrix.

  1. Follow up to Mentel test: Mantel tests are correlation tests that determine the correlation between two matrices (rather than two variables).  So, the positive correlation probably should be between the environmental parameter distance matrix and species abundance dissimilarity matrix created using a distance measure, in your case, Bray-curtis dissimilarity.

Response: Thank you to the reviewer for the valuable suggestions. We have included relevant explanations in the figure legends of the manuscript. The Mantel test's basic principle is to calculate the correlation coefficient (such as the Pearson correlation coefficient or the Spearman rank correlation coefficient) between two distance matrices. In the manuscript, we examined the relationship between environmental variables and species diversity by comparing the correlation between the environmental distance matrix and the biodiversity matrix. One matrix includes environmental parameters such as transparency, pH, dissolved oxygen, conductivity, total dissolved solids, and temperature. The other matrix includes diversity indices of the Salmo trutta fario,s gut microbiota under different farming modes, including "Observed","Chao1", "ACE", "Shannon", "Simpson", and "Fisher".

Reviewer 2 Report

Comments and Suggestions for Authors

Now your article looks much better.

Author Response

Reviewer #2

Now your article looks much better.

Response: Thank you for taking the time out of your busy schedule to review and comment on my manuscript. Your valuable feedback and suggestions have provided significant guidance. Your meticulous review and professional advice have not only enhanced the quality of this manuscript but have also given me clearer direction and standards for my future research. I am deeply grateful for the time and effort you have dedicated to this. I look forward to continuing to receive your invaluable guidance and support in the future.
